# CuInS_2_ Nanocrystals Embedded PMMA Composite Films: Adjustment of Polymer Molecule Weights and Application in Remote-Type White LEDs

**DOI:** 10.3390/nano13061085

**Published:** 2023-03-17

**Authors:** Qingchao Zhou, Zhongyi Shang

**Affiliations:** School of Jewelry, West Yunnan University of Applied Sciences, Tengchong 679100, China

**Keywords:** CuInS_2_ nanocrystals, PMMA, composite films, molecule weights, white LEDs

## Abstract

The commercial application of colloidal semiconductor nanocrystals has been realized owing to the development of composite film technology. Here, we demonstrated the fabrication of green and red emissive CuInS_2_ nanocrystals embedded polymer composite films of equal thickness by using a precise solution casting method. The impacts of polymer molecular weight on the dispersibility of CuInS_2_ nanocrystals were then systematically studied through evaluating the decrease in transmittance and red shift of emission wavelength. The composite films made from PMMA of small molecular weights exhibited higher transmittance. Applications of these green and red emissive composite films as color converters in remote-type light-emitting devices were further demonstrated.

## 1. Introduction

Colloidal semiconductor nanocrystals (NCs) are an emerging class of color-tunable light conversion materials, which have the advantages of solution processing, easy dispersion, tunable emission spectra, and high luminous efficiency [1,2,3,4,5]. Recently, NCs with excellent luminescence properties have been widely used in commercial display devices, such as CdSe NCs, InP NCs, and perovskite NCs [6,7,8,9]. To realize the application of colloidal semiconductor NCs, it is an essential step to disperse NCs in a matrix. Polymer materials, owing to their good processability and developed thin-film processes and industrial equipment that can be used directly, are the best matrices for NCs [10,11,12]. Therefore, dispersing NCs in polymer matrices to fabricate composite films is currently a hot research topic. The progress in NC-based composite films enables the configuration of light sources to change from “on-chip” to “on-surface” [13], thereby the influence of the heat released by chips on the NCs dispersed in films can be greatly reduced. As for lighting application, NC-based composite films can be integrated into remote-type LEDs to obtain planar light sources [14,15,16]. However, the dispersibility and stability of NCs in polymers are still the two key problems to be solved. The ligands on the surface of NCs are an important bridge between polymers and NCs [17], so the most reported research on NCs/polymer composites is the modification of the surface ligands to enhance the dispersibility and stability of NCs in polymers [18,19,20]. For example, Baek et al. fabricated the highly photo-stable composite films by embedding the thiol-capped perovskite NCs in a cyclic olefin copolymer [21].

CuInS_2_-based NCs are a new type of low-toxic phosphor material without heavy metals [22,23,24,25,26]. The potential of CuInS_2_-based NCs as phosphor-converted materials for white LED applications has been verified by Zhong et al. [27,28]. Dispersing CuInS_2_-based NCs in a polymer matrix needs to overcome many problems, among which the decrease in transmittance and red shift of emission wavelength due to the agglomeration of NCs and self-absorption are key issues to be evaluated. Especially for the application in white LED, the emission wavelength of NCs determines the CIE color coordinates of the fabricated devices.

Poly-methyl methacrylate (PMMA) is one of the most excellent organic optical materials generally used to make a variety of optical devices. Excellent transparency for visible light (~92%), good environmental inertness, better thermal stability, relatively low cost, easy processibility, etc. are considerable technologically useful properties of the PMMA film [29]. In this paper, PMMA was selected as the matrix of CuInS_2_-based NCs, and the CuInS_2_ NCs/PMMA composite films of equal thickness were prepared by a precise solution casting method. Then, the effect of PMMA molecular weight on the dispersibility and stability of CuInS_2_-based NCs was systematically studied by fluorescence emission spectroscopy and UV-Vis transmittance spectroscopy. The decrease in transmittance and the red shift in emission wavelength at 80 °C were further evaluated under a vacuum oven. Finally, the green and red emissive CuInS_2_ NCs/PMMA composite films were applied to a remote-type LED to achieve the uniform planar white light source. The adjustment of polymer molecular weight plays an important role in enhancing the optical properties of composite films. This article is a typical case of studying the dispersibility and stability of NCs in the polymer matrix with different molecular weights, which is an important research field where nanomaterials, interfaces, and composites intersect. It may also help to facilitate the commercial applications of nanocrystals from the perspective of polymer molecular design.

## 2. Materials and Methods

### 2.1. Materials

Copper(I) iodide (CuI, Alfa aesar, Haverhill, MA, USA, 98%), indium(III) acetate [In(OAc)_3_, Alfa aesar, Haverhill, MA, USA, 99.99%], zinc acetate dehydrate [(Zn(OAc)_2_, Aladdin, Shanghai, China 97%], 1-dodecanethiol (DDT, Alfa aesar, Haverhill, MA, USA, 98%), 1-octadecene (ODE, Alfa aesar, Haverhill, MA, USA, 90%), oleic acid (OA, Alfa aesar, Haverhill, MA, USA, 90%), oleylamine (OLA, J&K Scientific, Beijing, China, 90%), PMMA-15 k (J&K Scientific, Beijing, China), PMMA-350 k (Heowns, Tianjin, China), PMMA-550 k (Alfa aesar, Haverhill, MA, USA) were used without further purification.

### 2.2. Synthesis of CuInS_2_-Based NCs

Green and red emissive CuInS_2_-based NCs with emission peaks of 528 nm and 630 nm were prepared according to the method described by Chen et al. [22] A typical synthesis of the CuInS_2_-based NCs with a PL peak centered at 528 nm is as follows. Step I: CuI (0.19 g, 1 mmol), In(OAc)_3_ (1.16 g, 4 mmol), and Zn(OAc)_2_ (0.44 g, 2.5 mmol) were mixed with 10 mL DDT, 2 mL OA, and 20 mL ODE in a 100 mL three-necked flask. The mixture was heated to 220 °C and maintained for 15 min under nitrogen flow. Step II: 5 mL DDT was slowly injected into the as-prepared solution. An amount of 30 mL Zn stock solution [10.56 g Zn(OAc)_2_ in 10 mL OLA and 20 mL ODE] was then added drop-wise into the reaction mixture in 15 batches at intervals of 15 min. Step III: The resulting colloidal solution was cooled to room temperature and precipitated by adding excess acetone. The flocculant precipitate was centrifuged at 8000 rpm for 5 min, and the supernatant was decanted. Step II and step III were repeated twice. Step IV: The precipitate was then dispersed in a nonpolar solvent (toluene, chloroform). The washing process was repeated three times, and the precipitate was dried to a powder for further application. A typical synthesis of the CuInS_2_-based NCs with a PL peak centered at 630 nm is as follows. Step I: CuI (0.19 g, 1 mmol) and In(OAc)_3_ (1.16 g, 4 mmol) were mixed with DDT (5 mL), OA (2.5 mL), and ODE (25 mL) in a 100 mL three-necked flask. The solution was then heated to 220 °C to form a deep red colloidal solution. Step II: A fixed amount of Zn stock solution (2.64 g Zn(OAc)_2_, 10 mL OLA, and 10 mL ODE were, drop-by-drop, added into the reaction mixture in 10 batches at intervals of 15 min. Step III: The resulting colloidal solution was cooled to room temperature and precipitated by adding excess acetone. The flocculant precipitate was centrifuged at 8000 rpm for 5 min, and the supernatant was decanted. The washing process in step IV was the same as before.

### 2.3. Preparation of CuInS_2_ NCs/PMMA Composite Films

Step I: A transparent PMMA polymer solution was prepared by dissolving PMMA (5 g) in chloroform (40 mL) under ultrasound. Step II: A transparent CuInS_2_ NCs solution was prepared by dissolving CuInS_2_ NCs (0.0329 g) in chloroform (5 mL). Step III: An amount of 5 g of PMMA polymer solution and the above transparent CuInS_2_ NCs solution were mixed uniformly under ultrasound. Step IV: The mixed CuInS_2_ NCs/PMMA solution was transferred to an ultra-flat glass Petri dish. The solvent was naturally evaporated at room temperature for about 10 h. Then, the composite film embedded with CuInS_2_ NCs was peeled off the glass substrate and placed in a drying oven of 40 °C for 2 h until the solvent was completely evaporated. Green and red emissive CuInS_2_ NCs/PMMA composite films with different mass fractions were prepared by changing the weight of CuInS_2_ NCs, and the molecular weight of PMMA was added.

### 2.4. Characterizations of CuInS_2_ NCs/PMMA Composite Films

Ultraviolet-visible (UV-vis) transmittance spectra of the CuInS_2_/PMMA composite films were recorded on a UV-6100 spectrophotometer (Mapada Instruments Co., Ltd., Shanghai, China). The PL emission spectra of the CuInS_2_ based NCs dispersed in chloroform and CuInS_2_/PMMA composite films were taken using a FP-380 luminescence spectrometer (Gangdong Sci. & Tech. Development Co., Ltd., Tianjin, China). The absolute photoluminescence quantum yields (PLQYs) of composite films were determined using a PMA-12 calibrated multichannel spectrometer (Hamamatsu Photonics, Hamamatsu, Japan) with an C9920-02 integrated sphere (Hamamatsu Photonics, Hamamatsu, Japan). The luminance characteristics were obtained by a Photo Research PR680 spectroradiometer (Photo Research, Chatsworth, CA, USA). The CuInS_2_ NCs in chloroform and PMMA (ultrathin section samples) were analyzed using a JEOL-JEM 2100F TEM machine (JEOL, Tokyo, Japan) operating at an acceleration voltage of 200 kV. The ultrathin section samples were prepared using Leica EM UC7 ultramicrotome (Leica Microsystems, Wetzlar, Germany).

## 3. Results and Discussion

### 3.1. Solution Casting Method for CuInS_2_ NCs/PMMA Composite Films

Chloroform is a commonly used solvent for both PMMA and CuInS_2_ NCs, which was chosen for the fabrication of composite films. As illustrated in Figure 1, a typical solution casting method to fabricate CuInS_2_ NCs/PMMA composite films involved four stages. In stage I, green emissive CuInS_2_ NCs with an emission wavelength of 528 nm and red emissive CuInS_2_ NCs with an emission wavelength of 630 nm were separately dissolved in chloroform to prepare a transparent CuInS_2_ NCs solution. The TEM images of green and red emissive CuInS_2_ NCs were shown in Appendix A, and the average sizes of green and red emissive CuInS_2_ NCs are 5.6 ± 0.6 nm and 4.1 ± 0.2 nm, respectively. In stage II, the weight of the polymer solution and volume of CuInS_2_ NCs solution were calculated according to the mass concentration of the desired composite film. In stage III, when the solvent evaporated at room temperature, the as-fabricated composite film is very flexible owing to the small amount of solvent remaining in the film. One can mold the composite films into various shapes, such as the curved shape of a lampshade. In stage IV, the composite films will harden by further drying in an oven to remove the solvent completely. The composite films are difficult to separate from the mold if the solvent is completely evaporated. The equal thickness of composite films can be achieved by the precise weighing process, as well as by the fixing and leveling of the glass Petri dish mold.

### 3.2. Optical Properties of Green Emissive CuInS_2_ NCs/PMMA Composite Films

Four sheets of PMMA films of different molecular weights were prepared by adopting the same solution casting method, and the thicknesses were measured as 0.154 mm, 0.155 mm, 0.153 mm, and 0.154 mm respectively. The results verify the feasibility of preparing thin films with the same thickness by the solution casting method. Subsequently, we tested the transmittance spectra and fluorescence emission spectra of PMMA films with different molecular weights. It can be seen, from Appendix A, that the transmittances of PMMA films with different molecular weights are basically the same in the visible light range (~92%), and there is no obvious fluorescence emission in the visible light range.

Transmittance and emission wavelength are the two most important optical parameters to be evaluated for light conversion films. We have quantitatively analyzed the correlation between the mass concentration of CuInS_2_-based NCs in PMMA and the emission wavelength. The emission wavelength of the composite film red-shifted as the mass concentration of CuInS_2_ based NCs increases. The emission wavelength of the composite film is 541 nm when the mass concentration of CuInS_2_ based NCs reaches 10%. Compared with the emission wavelength of CuInS_2_ NCs toluene solution (559 nm) with equal mass concentration, it is obvious that the red shift of CuInS_2_ NCs in PMMA is relatively smaller. Therefore, the red shift of CuInS_2_-based NCs caused by the agglomeration and self-absorption can be effectively reduced by dispersing them into the polymer matrix.

In order to better reveal the optical properties (transmittance and emission wavelength) of the CuInS_2_ NCs/PMMA composite films, PMMA of four different molecular weights was used for comparison experiments. Figure 2 provides an overview of sixteen composite films prepared by dispersing green emissive CuInS_2_ NCs in PMMA with different molecular weights. From left to right, the mass concentration of CuInS_2_ NCs increases, and from top to bottom, the molecular weight of PMMA increases. As the mass concentration of CuInS_2_ NCs increases, the transmittance of the as-fabricated composite film decreases, and the appearance color of the composite film changes from light green to yellow.

It is difficult to evaluate the dispersibility of CuInS_2_ NCs in PMMA with different molecular weights by the naked eyes. Therefore, we tested the transmittance and fluorescence spectra of these composite films. Figure 3a,b show the transmittance spectra of CuInS_2_ NCs/PMMA composite films with a mass concentration of 1% and 2%, respectively. It is obvious that the dispersion of CuInS_2_ NCs is the best in PMMA with a molecular weight of 15 k, since the composite film has the highest transmittance. Figure 3c,d show the fluorescence spectra of CuInS_2_ NCs/PMMA composite films with a mass concentration of 1% and 2%, respectively. The emission spectra are partly enlarged to demonstrate the red shift of the emission wavelength more clearly. It can be seen from the figure that the red-shift of emission wavelength is the smallest when the CuInS_2_ NCs were dispersed in PMMA with a molecular weight of 15 k. Combined with the light transmittance and emission wavelength of the composite films, we believe that green emissive CuInS_2_ NCs have the best dispersion in the PMMA of 15 k. The poor dispersion of CuInS_2_ NCs in the polymer matrix will cause a decrease in light transmittance and a red-shift of emission wavelength. In low-molecular-weight polymers, the molecular chains of the polymers are relatively free to move, and the entropy changes required for polymer penetration are usually low [30,31,32,33]. Therefore, the molecular chains penetrate easily with the short-chain ligands (OLA) on the surface of CuInS_2_ NCs to establish the interface equilibrium.

From the data in Figure 2, it is apparent that the CuInS_2_ NCs with high mass concentration can still be uniformly distributed in the PMMA matrix, although the light transmittance is low. The CuInS_2_ NCs dispersed in PMMA can maintain ~80% of the original PLQYs. The ordered agglomeration of CuInS_2_ NCs in PMMA will not cause serious fluorescence quenching owing to the slow solvent evaporation at room temperature [34,35].

### 3.3. Optical Properties of Red Emissive CuInS_2_ NCs/PMMA Composite Films

The dispersibility of green emissive CuInS_2_ based NCs in PMMA is obviously affected by the molecular weight of PMMA. However, the light transmittance of the composite film is not high, even though the mass concentration of CuInS_2_ NCs is very low (1%), which will increase the error of quantitative analysis. In this part, the red emissive CuInS_2_ NCs/PMMA composite films with high transmittance were applied to analyze the effect of molecular weight on optical properties.

Appendix A shows the red emissive CuInS_2_ NCs/PMMA composite films prepared by using the same solution casting method. It can be seen from Appendix A that the transmittance of the as-fabricated composite films exceeds 60% when the mass concentration of red emissive CuInS_2_ NCs in PMMA reached 5%. However, the transmittance of the green emissive CuInS_2_ NCs/PMMA composite film with the same mass concentration is only 5%. We attributed this improvement in transmittance to the quality of ligands on the surface of CuInS_2_ NCs. The red emissive CuInS_2_ NCs were only covered once by a ZnS shell during the synthesis process, but the green emissive CuInS_2_ NCs were covered by ZnS shells three times during the synthesis process. It should be noted that the CuInS_2_ NCs need to be cleaned before each covering process to remove the (organic) residuals and by-products from the surface of NCs, so the number of ligands on the surface will decrease, which has been confirmed by Akdas et al. by using nuclear magnetic resonance spectroscopy [36]. The reducing of surface ligands leads to the weak dispersibility of CuInS_2_ NCs within the polymer matrix.

Looking at Appendix A, it is apparent that the red emissive composite film made from low molecular weight PMMA (15 k) has the highest light transmittance. Instead, the composite film made from high molecular weight PMMA (550 k) has the lowest light transmittance, which is consistent with the results of green emissive CuInS_2_ NCs/PMMA composite films. The thickness of the four red emissive composite films was measured as 0.156 mm, 0.153 mm, 0.154 mm, and 0.155 mm, respectively. Therefore, the effect caused by thickness can be excluded.

In addition, two molecular weights of PMMA were blended as the polymer matrix to further reveal the influence of molecular weight on the optical properties of composite films. The thickness of the as-fabricated composite film was measured as 0.158 mm, 0.156 mm, 0.155 mm, 0.155 mm, and 0.156 mm, respectively. Molecular weights of 550 k and 15 k were selected because their corresponding composite films have a maximum and a minimum transmittance. The mass concentration of red emissive CuInS_2_ NCs in PMMA was set to 20%. Two molecular weights of PMMA were mixed in volume ratios of 1:4, 1:1, and 4:1, respectively. It can be seen, from Figure 4, that the composite film prepared with 550 k PMMA still has the lowest transmittance. However, the light transmittance of the composite film prepared with blended PMMA (550 k/15 k = 1/4) is the highest. It can be concluded that the hybrid of PMMA with different molecular weights can improve the dispersion of CuInS_2_ NCs in PMMA. Based on the “Bimodal Surface Ligand Engineering” of nanocrystals reported by Li et al. [37], we explain this phenomenon with a “Bimodal Molecular Weight Strategy” for the case that CuInS_2_ NCs were dispersed within a polymer matrix composed of two molecular weights. On the one hand, the polymer chains with low-molecular-weights can more easily interpenetrate with the organic ligands on the surface of CuInS_2_ NCs to guarantee the interfacial compatibility of NCs and polymers; on the other hand, the polymer chains with high-molecular-weights can prevent the agglomeration of CuInS_2_ NCs by inhibiting their free movements.

### 3.4. Dispersibility of CuInS_2_ NCs in Green and Red Emissive Composite Films

To further analyze the reasons for the differences in transmittance and red-shift of composite films, in this paper, the CuInS_2_ NCs/PMMA composite films were further characterized by TEM. The ultrathin section samples for TEM test were prepared using Leica EM UC7 ultramicrotome. Figure 5a,b are the TEM images of green and red emissive CuInS_2_ NCs in PMMA matrix respectively. It is obvious that the green emissive CuInS_2_ NCs underwent more severe agglomeration in the PMMA matrix, and the size of the aggregates in PMMA matrix is 10–30 nm. In contrast, the red emissive CuInS_2_ NCs maintain a good monodispersity in the PMMA matrix, and the size of the CuInS_2_ NCs in the PMMA matrix is around 4 nm.

### 3.5. Vacuum Thermal Stability of Red Emissive CuInS_2_ NCs/PMMA Composite Films

The effect of temperature on the dispersion of CuInS_2_ NCs in PMMA was evaluated by vacuum heating. The use of a vacuum drying oven can prevent the composite films from being affected by oxygen and water. In the beginning, the red emissive CuInS_2_ NCs/PMMA composite films with different molecular weights were placed in a vacuum drying oven at 80 °C for 5 h, and the light transmittance and fluorescence spectra were recorded. It can be seen, from Figure 6a, that the transmittance of the composite films decreased greatly. After that, the composite films were placed into the vacuum drying oven at 80 °C again for another 25 h, and the light transmittance and fluorescence spectra were recorded. It can be seen from the green line in the figure that the decrease in the light transmittance of the film is not as serious as before, which means the spatial distribution of CuInS_2_ NCs within the PMMA gradually tends to be stable during the long-term heating process. After heating for 30 h, the transmittance of the composite film prepared with pure 15 k PMMA decreased by 44.9%, but the one prepared with pure 550 k PMMA decreased by only 26.4%. The decreasing transmittance of the composite films decreased with the increase in the concentrations of 550 k PMMA, which were 40.9%, 33.8%, and 30.5%, respectively. The specific transmittances and emission wavelengths are summarized in Table 1.

It can be seen from Figure 6b that the fluorescence emission peaks of the composite films have a slight red-shift after heating in the vacuum-drying oven for 30 h. The red-shift of the composite film prepared with pure 550 k PMMA is the smallest (4.8 nm). This can be attributed to the difficult displacement of PMMA chains with large molecular weights, so the agglomeration of CuInS_2_ NCs dispersed in the PMMA is not easy to occur.

### 3.6. Application of CuInS_2_ NCs/PMMA Composite Film in Remote White LEDs

The emergence of the remote-type LED greatly improves the possibility of CuInS_2_ NCs/PMMA composite film being applied in white LED. Figure 7a shows the schematic diagram of the white LED with remote structure by combining the blue LED chips and the dual-layer CuInS_2_/PMMA composite films. The surface of the reflector cup in this configuration is diffuse, and the reflected light has a Lambertian distribution for a diffuse reflector cup [38]. In this paper, a blue light chip (emission peak at 455 nm) of 10 W was used as the excitation light source, and a combination of 20 wt% green emissive CuInS_2_ NCs/PMMA composite film and 5 wt% red emissive CuInS_2_ NCs/PMMA composite film were selected to prepare high-quality white light. As shown in Figure 7b, the emission spectrum covers a very wide wavelength range, the luminance of the white light is 621 lm, the color coordinate of the white light is (0.33, 0.34), the color temperature is 5608 K, and the color rendering index Ra is 90. The luminance of the original blue LED before integrating the dual-layer CuInS_2_/PMMA composite films is 402 lm. High-quality white LEDs can be obtained by using CuInS_2_ NCs/PMMA composite films of two wavelengths, which are attributed to the relatively broad emission spectra of CuInS_2_ NCs. In addition, embedding the CuInS_2_ NCs of two emission wavelengths into two layers of composite films is beneficial to improve the luminous efficiency of white LEDs, since the serious mutual absorption among CuInS_2_ NCs of different emissions can be avoided. That is, the green light excited by the blue light will be absorbed by the red emissive CuInS_2_ NCs.

## 4. Conclusions

In this paper, the green and red emissive CuInS_2_ NCs embedded polymer composite films of equal thickness were fabricated by using a precise solution casting method. The UV-Vis transmittance spectra and emission spectra of all the CuInS_2_ NCs/PMMA composite films were measured and analyzed. The experimental results show that the transmittance of the composite films is the highest, and the corresponding red shift of the CuInS_2_ NCs is the smallest when the molecular weight of PMMA is 15 k. As the mass concentration of green emissive CuInS_2_ NCs exceeds 5%, the transmittance of the composite film decreases to a very low level, but the film still maintains high PLQYs due to the ordered agglomeration. From the viewpoint of improving the dispersibility (transmittance) of CuInS_2_ NCs in the polymer, it is better to select a polymer matrix with small molecular weights, but a polymer matrix with large molecular weights exhibits relatively better vacuum thermal stability. The blending of PMMA with different molecular weights as the matrix of the composite films can simultaneously improve the dispersibility and thermal stability of CuInS_2_ NCs. High-quality planar white LEDs with color coordinates of (0.33, 0.34), color temperature of 5608 K, and color rendering index of 90 have been realized by simultaneously integrating the red and green emissive CuInS_2_ NCs/PMMA composite films to the remote-type LED modules. More importantly, the versatility of this molecular designing strategy would open up a door to enhance the dispersibility and stability of functional NCs within PMMA or other polymers.

## Figures and Tables

**Figure 1 nanomaterials-13-01085-f001:**
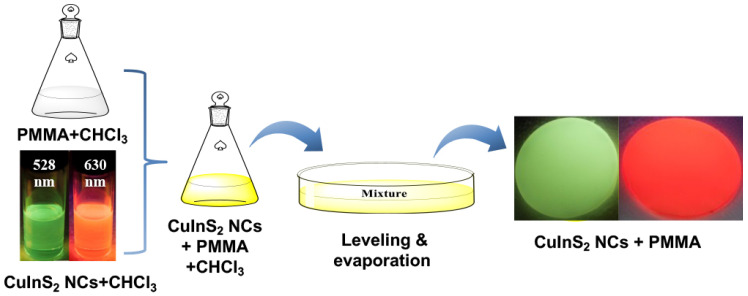
Schematic illustration of the fabrication of CuInS_2_ NC-embedded PMMA composite films.

**Figure 2 nanomaterials-13-01085-f002:**
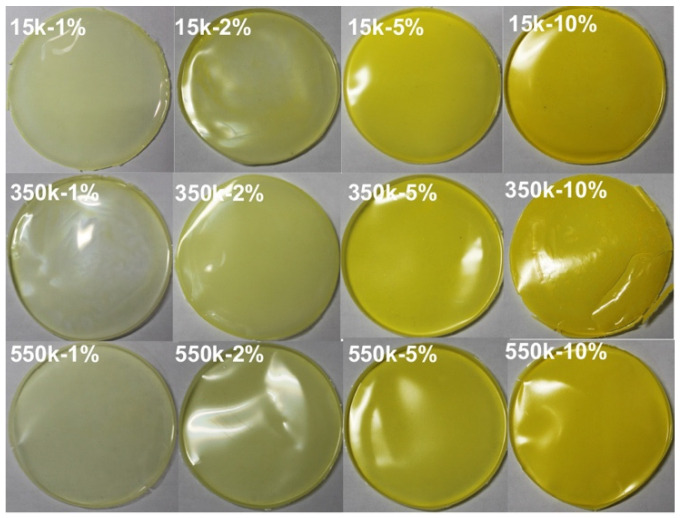
Green emissive CuInS_2_ NCs/PMMA composite films with different mass concentration and PMMA molecular weight. From left to right, the mass concentrations are 1 wt%, 2 wt%, 5 wt%, and 10 wt%, respectively.

**Figure 3 nanomaterials-13-01085-f003:**
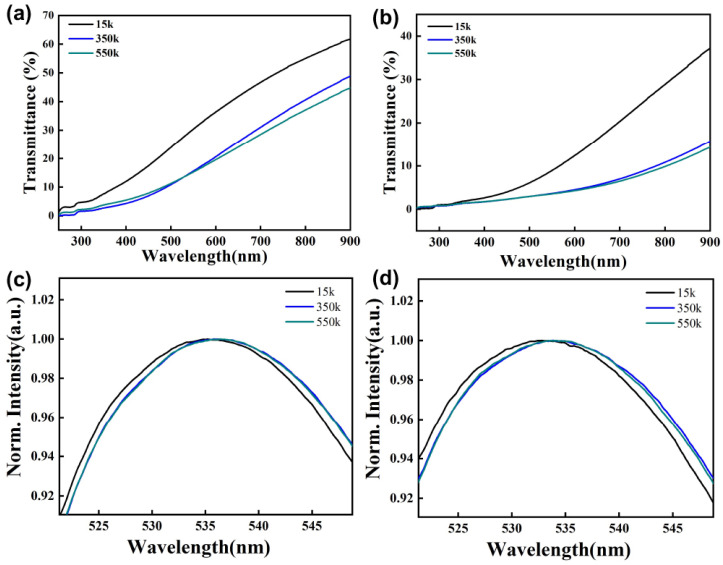
Transmittance and fluorescence spectra of CuInS_2_ NCs dispersed in PMMA of different molecular weights. (**a**,**c**) The mass concentration is 1%; (**b**,**d**) The mass concentration is 2%.

**Figure 4 nanomaterials-13-01085-f004:**
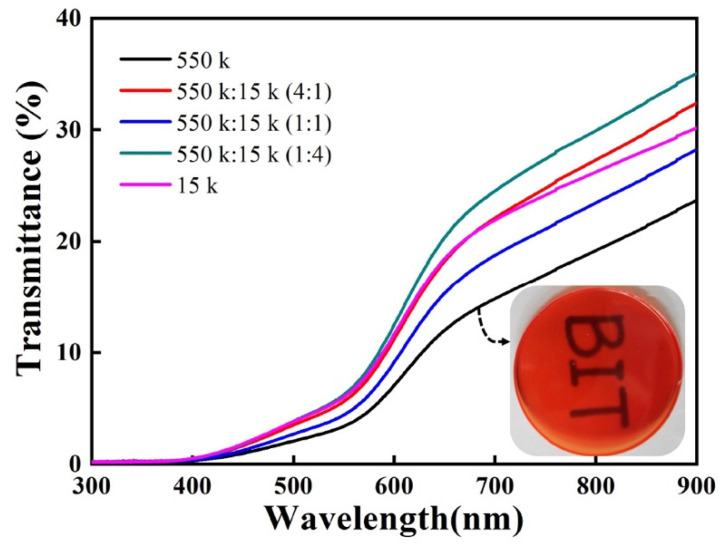
Transmittance spectra of red emissive composite films (20 wt% CuInS_2_ NCs) prepared by blending PMMA with different molecular weights.

**Figure 5 nanomaterials-13-01085-f005:**
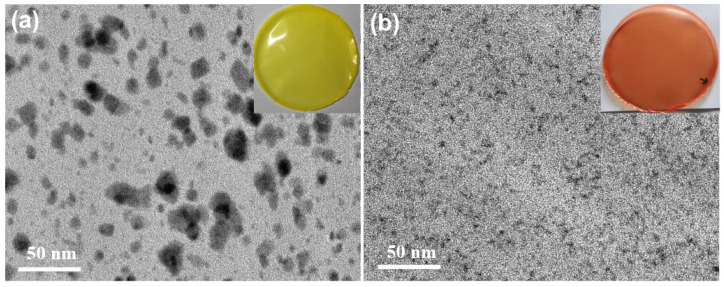
(**a**) TEM images of green emissive CuInS_2_ NCs in PMMA matrix; (**b**) TEM images of red emissive CuInS_2_ NCs in PMMA matrix.

**Figure 6 nanomaterials-13-01085-f006:**
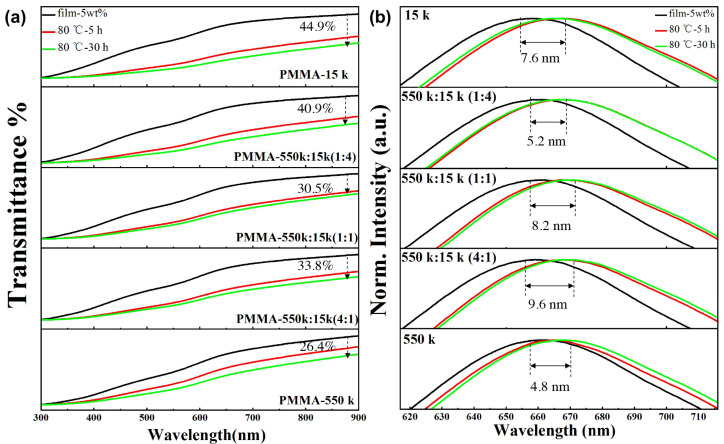
Spectral changes of red emissive composite films (5 wt% CuInS_2_ NCs) prepared by blending PMMA with molecular weights of 550 k and 15 k. (**a**) The transmittance spectra before and after heating at 80 °C; (**b**) The fluorescence emission spectra before and after heating at 80 °C.

**Figure 7 nanomaterials-13-01085-f007:**
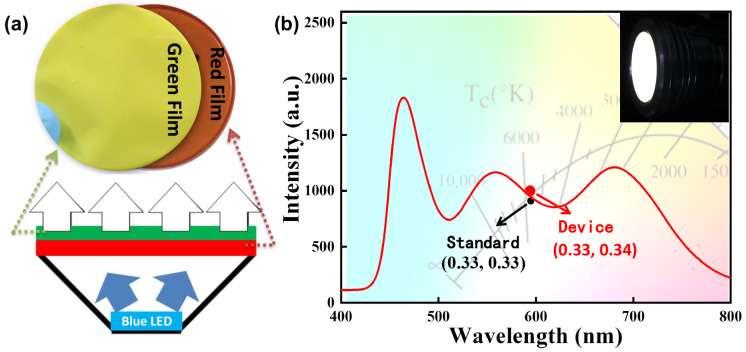
(**a**) Schematic diagram of the white light LED with remote structure by combinating the blue LED chip and the dual-layer CuInS_2_/PMMA composite film; (**b**) the white light spectrum and the corresponding color coordinate of remote white LED in the inset.

**Table 1 nanomaterials-13-01085-t001:** Transmittances and emission wavelengths of the composite films before and after heat treatment under vacuum drying oven.

M.W. of PMMA	T%	T% @ 30 h	WL	WL @ 30 h	Red-Shift
15 k	76.25	42.01	658.8 nm	666.4 nm	7.6 nm
550 k/15 k 1:4	75.68	44.69	661.4 nm	666.6 nm	5.2 nm
550 k/15 k 1:1	73.58	51.12	661.2 nm	669.4 nm	8.2 nm
550 k/15 k 4:1	73.52	48.66	659.2 nm	668.8 nm	9.6 nm
550 k	68.67	50.53	662.4 nm	667.2 nm	4.8 nm

## Data Availability

Data are available from the corresponding author.

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
