# Peer review of "CuInS2 Nanocrystals Embedded PMMA Composite Films: Adjustment of Polymer Molecule Weights and Application in Remote-Type White LEDs"

_nanomaterials, 2023, doi:10.3390/nano13061085_

Round 1

Reviewer 1 Report

The authors describe a research article entitled “CuInS2 Nanocrystals Embedded PMMA Composite Films: Adjustment of Polymer Molecule Weights and Application in Remote-Type White LEDs”. The topic of the manuscript is interesting, and the manuscript constitutes an interesting article concerning the development of filters for producing white light with CIE coordinates close to the perfect white light (0.33, 0.33). A short conclusion highlighting the main results of this work. No perspectives to this work are given at the end of the document.

The work is well-written and a well-constructed introduction has been established by the authors. Sufficient spectra and figures are included in the manuscript for comprehension and clarity. Numerous figures in colors have been introduced in the manuscript, rendering the article more attractive. Interesting and convincing results are also presented in this manuscript. Overall, I think that this is a manuscript that I recommend for publication after inclusion of minor revisions.

1) What about the mechanical properties of these PMMA films ? Are these films flexible or easy to break ?

2) The authors mentioned that a LED emitting at 405 nm and with a light intensity of 10 W was used. What about the luminance before and after the PMMA filter ? In fact, the filters are strongly colored and even if a white light is produced, most of it is certainly absorbed by the glass filter.

3) What about the distribution of light ? Does the white light exhibit a Lambertian distribution ?

4) Perspectives should be added to this work.

Reviewer 2 Report

Referee report

The article “CuInS2 Nanocrystals Embedded PMMA Composite Films: Ad-2 justment of Polymer Molecule Weights and Application in 3 Remote-Type White LEDs” is devoted to developing of white LEDs based on organic / not organic nanocrystals composites. It is quite actual and interesting topic. The authors proposed an original approach, I would like to note the ease of obtaining a composite material. Composite materials are very relevant and promising, for example, based on graphene, see Visualization of Swift Ion Tracks in Suspended Local Diamondized Few-Layer Graphene” Materials, 16, 1391 (2023). DOI: https://doi.org/10.3390/ma16041391. I would like to wish the authors to make LED with electric pumping and not with optical pumping! The article contains new experimental data and will be interesting for researchers and technologists. The article can be published. Below are some comments that authors can take into account.

Comment:

1) The work is good written, but it is necessary to explain all the abbreviations used - for example DDT, OA, OLA.

2) As advice for the future, I would like to wish the authors to use a very powerful and informative Raman technique for the analysis of nanocrystals. It is known that this technique can also be used to study the sizes of nanocrystals, for example, using Improved Model of Optical Phonon Confinement, see Journal of Experimental and Theoretical Physics, 2013, Vol. 116, No. 1, pp. 87–94 DOI: 10.1134/S1063776112130183.

Accept.

Reviewer 3 Report

Dear Authors,

The manuscript submitted for review, entitled "CuInS2 Nanocrystals Embedded PMMA Composite Films: Adjustment of Polymer Molecule Weights and Application in Remote-Type White LEDs," by Qingchao Zhou and Zhongyi Shang of West Yunnan University of Applied Sciences, Tengchong, concerns the fabrication of green and red emission CuInS2 nanocrystals embedded in polymer composite films of equal thickness using a solution casting method.

In general, the work is interesting, but needs major revisions to make it more attractive and adapt it to the requirements of scientific papers.

Thus, the comments are as follows:

1. In the introduction of the manuscript, the scientific purpose of the work carried out should be formulated.

2. In the introduction, the authors should refer a little more broadly to the whole problem and not only to the CuInS2 37-based NC dispersion in a polymer matrix and briefly present the results obtained by other researchers.

3. The authors should highlight what the novelty and originality of their research is and what their results contribute to the field they represent.

4. Section 3.1 does not constitute results or discussion and as such should be moved to section 2.

5. The discussion conducted is very limited and amounts to a direct interpretation of the results obtained. However, there should be an attempt to explain why this is the case and support your arguments with the results achieved by other scientists. In this section on the analysis of the results obtained, more reference should be made to the source literature, which will confirm the cited formulations and justifications of the authors.

After incorporating these corrections, the manuscript can be published in the journal Nanomaterials.

Best regards,

Round 2

Reviewer 3 Report

Dear Authors,

Thank you very much to the Authors for taking my remarks and comments into account. Thanks to this, the content of the manuscript and scientific reliability have increased significantly. All my comments have been taken into account in an appropriate and proper manner, requiring no further explanation or supplementation.

The manuscript can be printed in the journal Nanomaterials.

Best regards,